# Efficacy of Respiratory Syncytial Virus Vaccination to Prevent Lower Respiratory Tract Illness in Older Adults: A Systematic Review and Meta-Analysis of Randomized Controlled Trials

**DOI:** 10.3390/vaccines12050500

**Published:** 2024-05-05

**Authors:** Matteo Riccò, Antonio Cascio, Silvia Corrado, Marco Bottazzoli, Federico Marchesi, Renata Gili, Pasquale Gianluca Giuri, Davide Gori, Paolo Manzoni

**Affiliations:** 1AUSL–IRCCS di Reggio Emilia, Servizio di Prevenzione e Sicurezza Negli Ambienti di Lavoro (SPSAL), Local Health Unit of Reggio Emilia, 42122 Reggio Emilia, Italy; 2Infectious and Tropical Diseases Unit, Department of Health Promotion, Mother and Child Care, Internal Medicine and Medical Specialties, “G D’Alessandro”, University of Palermo, AOUP P. Giaccone, 90127 Palermo, Italy; antonio.cascio03@unipa.it; 3ASST Rhodense, Dipartimento della donna e Area Materno-Infantile, UOC Pediatria, 20024 Milano, Italy; 4Department of Otorhinolaryngology, APSS Trento, 38122 Trento, Italy; 5Department of Medicine and Surgery, University of Parma, 43126 Parma, Italy; 6Department of Prevention, Turin Local Health Authority, 10125 Torino, Italy; 7Department of Medicine and Diagnostics, AUSL di Parma, 43100 Parma, Italy; 8Department of Biomedical and Neuromotor Sciences, University of Bologna, 40126 Bologna, Italy; davide.gori4@unibo.it; 9Department of Public Health and Pediatric Sciences, University of Torino School of Medicine, 10125 Turin, Italy; p.manzoni@unito.it

**Keywords:** RSV, viral pneumonia, vaccine, immunization

## Abstract

A systematic review and meta-analysis was designed in order to ascertain the effectiveness of respiratory syncytial virus (RSV) vaccination in preventing lower respiratory tract diseases (LRTD) in older adults (age ≥ 60 years). Studies reporting on randomized controlled trials (RCTs) were searched for in three databases (PubMed, Embase, and Scopus) and the preprint repository medRxiv until 31 March 2024. A total of nine studies were eventually included, two of which were conference proceedings. Our analysis included five RCTs on five RSV vaccines (RSVpreF, RSVPreF3, Ad26.RSV.preF, MEDI7510, and mRNA-1345). The meta-analysis documented a pooled vaccine efficacy of 81.38% (95% confidence interval (95% CI) 70.94 to 88.06) for prevention of LRTD with three or more signs/symptoms during the first RSV season after the delivery of the vaccine. Follow-up data were available for RSVPreF3 (2 RSV seasons), RSVpreF (mid-term estimates of second RSV season), and mRNA-1345 (12 months after the delivery of the primer), with a pooled VE of 61.15% (95% CI 45.29 to 72.40). After the first season, the overall risk for developing RSV-related LRTD was therefore substantially increased (risk ratio (RR) 4.326, 95% CI 2.415; 7.748). However, all estimates were affected by substantial heterogeneity, as suggested by the 95% CI of I2 statistics, which could be explained by inconsistencies in the design of the parent studies, particularly when dealing with case definition. In conclusion, adult RSV vaccination was quite effective in preventing LRTD in older adults, but the overall efficacy rapidly decreased in the second season after the delivery of the vaccine. Because of the heterogenous design of the parent studies, further analyses are required before tailoring specific public health interventions.

## 1. Introduction

Human respiratory syncytial virus (RSV) is a pleomorphic, enveloped, single-stranded negative-sense RNA virus (15 to 16 kb) of medium size (120–300 nm diameter) [1,2,3,4]. RSV belongs to the order of mononegavirales and the genus orthopneumovirus (family *Pneumoviridae*) [4]. RSV is usually acknowledged as a highly contagious pathogen [5,6]: before 2020 and the COVID-19 pandemic, RSV was by far the single most common viral cause of acute respiratory infections (ARIs) and lower respiratory tract disease (LRTD) [3], accounting for around 33 million cases every year [2,7]. Until recently, cases occurred during seasonal epidemics that, in the northern hemisphere, extensively overlap with other respiratory viruses such as influenza, adenovirus, and SARS-CoV-2 [8,9]. In countries with temperate climates, seasonal epidemics occurred during the winter season, peaking between December and January [2,7], while in tropical countries, RSV outbreaks are associated with the summer season because of the hot, humid, and rainy climate [10,11,12]. 

According to available estimates, every year RSV causes around 3.5 million hospital admissions in infants aged 0 to 60 months [1,3], even without noticeable comorbidities [2,13,14], with high hospital admission rates [15,16,17,18,19], recently estimated at around 5.3 per 1000 people (95% confidence interval (95% CI)) and 4.2–6.8 at a global level [1,3]. Nonetheless, with around 158,229 hospitalizations occurring annually in adults aged ≥18 years, 92% of which are from adults aged ≥65 years [20], with a corresponding hospitalization rate of 157 per 100,000 [21], RSV also affects older individuals [13,22,23,24]. It is increasingly acknowledged as a cause of respiratory illnesses and severe lower respiratory tract infections in adults [10], mostly older adults [25,26], in whom it causes high morbidity and mortality [24,27,28], particularly among institutionalized subjects [2,29,30,31], resulting in a significant public health impact [32,33,34,35,36,37]. Until 2023, preventive and treatment options for older adults were limited, as several RSV candidate vaccines ultimately failed to achieve targeted efficacy. In children, however, these disease candidates were initially identified as having a greater than 70% vaccine efficacy (VE) against confirmed severe RSV disease over at least one year post vaccination, as well as the secondary target of 50% VE against confirmed severe RSV [38]. Since then, several vaccines have been able to complete phase III clinical trials, and two preventive vaccines were ultimately licensed for human use: Abrysvo from Pfizer Inc. (Pfizer Europe MA EEIG, Brussels, Belgium) and Arexvy from GlaxoSmithKline LLC (GlaxoSmithKline Biologicals SA, Rixensart, Belgium) [39,40]. Both of them are subunit vaccines based on pre-fusion F protein, but while Abrysvo is a divalent, non-adjuvated formulate, Arexvy is a monovalent formulate based on the pre-fusion protein F of RSV from group A [39,40]. Their overall efficacy in older adults has been documented in specifically designed randomized controlled trials (RCTs) [41,42], leading to official recommendations for their use in individuals aged ≥60 years [43]. However, other RSV vaccines are or have been in the pipeline for use in older adults [5,44,45,46,47,48,49,50,51,52,53,54,55,56,57,58,59,60,61,62,63,64]. Even though some high-quality systematic reviews on the VE of RSV vaccines in pregnant women have been published [65], to the best of our knowledge no studies have to date tried to summarize the available evidence on RSV vaccines for older adults. Therefore, this study was designed in order to systematically evaluate the VE of RSV vaccines in older adults, providing valuable evidence and possibly guidance for clinical use.

## 2. Materials and Methods

### 2.1. Research Concept

This systematic review with meta-analysis was designed in accordance with the “Preferred Reporting Items for Systematic Reviews and Meta-Analysis” (PRISMA) statement [66] (see Appendix A). Prior to conducting the study, it was registered in the PROSPERO database (progressive registration number CRD42023475548 https://www.crd.york.ac.uk/prospero/, accessed on 21 March 2024) [67]. 

Research concepts were defined by means of the “PECO” strategy (i.e., patient/population/problem, exposure, control/comparator, outcome) [68,69] (Table A1). More precisely, the assessment concerned whether adults aged 60 years or older having received at least one dose of any RSV vaccine (P), upon exposure to RSV infection during the subsequent RSV season (E), incurred LRTD or ARI (O) in reduced occurrence compared to adults aged 60 years or more who were not vaccinated against RSV (placebo) (C).

### 2.2. Research Strategy 

The systematic retrieval was performed across three databases (Pubmed; EMBASE; and Scopus) and the preprint repository medRxiv from their inception until 31 March 2024 (Table A2). The search strategy was obtained by modifying the blueprint originally recommended by Ma et al. [65] for maternal RSV vaccine studies and resulted in the combination of specifically designed search strings provided in Table A2. Where allowed by the searched database, all inquiries were restrictive to individuals aged 60 years or older. 

### 2.3. Selection Criteria

The criteria for being included in the retrieved studies were as follows:

(1) Vaccination of older adults (age ≥ 60 years);

(2) RSV vaccine administration (any);

(3) Comparison of RSV vaccine efficacy with placebo in a randomized controlled trial (RCT);

(4) Reporting on the efficacy of the RSV vaccine in terms of ARI or LRTD. 

The following exclusion criteria were then applied:

(1) Vaccination of children and/or younger adults (age < 60 years);

(2) Secondary studies (i.e., systematic reviews, meta-analysis, letters, editorial comments, case reports, preprint from repositories other than medRxiv);

(3) Studies not designed as RCTs;

(4) Studies on animals (including non-human primates) or preclinical testing;

(5) Outcomes other than clinical efficacy;

(6) full text not available either through online repositories or through inter-library loan, or the main text written in a language different from English, Italian, German, French, Spanish, or Portuguese;

(7) lack of details about geographical setting and corresponding time frame;

(8) laboratory diagnosis of respiratory infections performed with methods other than real-time quantitative polymerase chain reaction (RT-qPCR) or non-RT-qPCR nucleic acid amplification tests (NAATs).

In case of cross-publication and/or duplicated series, only the most recent publication was included, and if it proved to be feasible, duplicated data were removed from both qualitative and quantitative analysis.

### 2.4. Selection Criteria

By using inclusion and exclusion criteria, retained articles were then screened to ascertain their relevance for the research question. The articles were initially title screened using the title alone and all the items were positively assessed; hence, they were screened by reading the whole abstract [66,70]. All the retained entries were eventually full-text screened and independently rated by two investigators (S.C. and F.M.). Disagreements were resolved primarily by consensus between investigators and, when it was not reached, by involving as a third person the chief investigator (M.R.). 

### 2.5. Data Extraction

The extracted data were retrieved from both the main text and the Appendix A (where available) and included:

(a) Main characteristics of the study: first author’s name, year of publication, RCT registration number, assessed vaccine;

(b) Characteristics of the RCT: sample size of the study groups (recipients of vaccine and placebo), baseline data (demographics and comorbidities), number of countries involved in the study, timeframe of the study;

(c) Outcome data: definition of primary and secondary outcome; case definition. The main outcome encompassed the reported ARI and LRTD.

### 2.6. Quality Assessment (Risk of Bias)

To cope with potential risk of bias (ROB) due to research practices [71,72,73], the ROB tool from the National Toxicology Program (NTP)’s Office of Health Assessment and Translation (OHAT) (now the Health Assessment and Translation (HAT) group) [73,74] was used. According to its original design, the OHAT ROB evaluates the internal validity of a given study by weighting 6 potential sources of bias: participant selection (D1), confounding factors (D2), attrition/exclusion (D3), detection (D4), selective reporting (D5), and other sources of bias (D6). All sources of bias are rated on a 4-point scale ranging from “definitely low,” “probably low,” “probably high,” to “definitely high.” The OHAT ROB was prioritized over other similar instruments, as it does not provide an overall rating for each study nor require that studies affected by a certain degree of ROB be removed from the pooled analyses [74].

### 2.7. Data Analysis 

As a preliminary step, individual estimates for vaccine efficacy (VE) were calculated for each study. VE can be defined as the percentage reduction of disease cases in a vaccinated group of people compared to an unvaccinated group. Mathematically, VE was defined as:VE = (1 − Relative Risk) × 100% (1)

Meta-analysis of demographic data from individual studies was performed through calculation of the risk ratio (RR) as an effect index. Similarly, comparison of breakthrough infections in follow-up studies was performed by calculation of the RR. Pooled estimates of RR and VE were calculated through a random-effects model (REM) meta-analysis of retrieved studies and reported as point estimates with their 95% confidence intervals (95% CIs). As the search strategy likely included both primary studies (i.e., studies performed on subjects never vaccinated against RSV in their first post-vaccine RSV season) and follow-up studies (i.e., studies performed on subjects previously vaccinated, after the first RSV season), two distinctive pooled VE estimates were calculated for primary and follow-up studies, respectively. The REM was preferred over a fixed-effects model, as it is considered more effective in dealing with the genuine differences underlying the results of the studies (heterogeneity) [75,76]. 

The heterogeneity of the literature (i.e., the inconsistency of the effects between the included studies) was defined as the percentage of total variation across studies likely due to heterogeneity rather than chance [71], and was quantified by means of the I^2^ statistic through the following categories: 0 to 25%, low heterogeneity; 26% to 50%, moderate heterogeneity; and ≥50%, substantial heterogeneity. Because of the presumptive small size of the meta-analyses, with likely underestimation of actual heterogeneity by point estimate of I^2^, 95% CIs were provided [71]. 

Sensitivity analysis (i.e., the study of how the uncertainty in the output of a mathematical model or system can be apportioned to different sources of uncertainty in its inputs) was performed to evaluate the effect of each study on the pooled estimates by excluding one study at a time. 

Publication bias was visually assessed through the calculation of contour-enhanced funnel plots, and their asymmetry was eventually assessed by means of the Egger’s test for all outcomes with three or more included studies [66,77]. Small-study bias was assessed by generating corresponding radial plots. A *p*-value < 0.05 was considered statistically significant for both publication and small-study bias.

All calculations were performed by means of R (version 4.3.1) [78] and Rstudio (version 2023.06.0 Build 421; Rstudio, PBC; Boston, MA, USA) software by means of the packages meta (version 7.0) and fmsb (version 0.7.5). The Prisma2020 flow diagram was designed by means of the PRISMA2020 package [79].

## 3. Results

### 3.1. Descriptive Analysis

As shown by the flowchart of the search and selection process (Figure 1), a total of 2096 entries were retrieved, including 432 studies from PubMed (20.61%), 344 from EMBASE (16.41%), 721 from Scopus (34.40%), and 599 from MedRxiv (28.58%). After the removal of duplicates, 312 records were screened for title and abstract (14.89% of the original pool). 

After the removal of duplicates and the full-text screening, nine articles were included in the pooled analysis, from five RCTs (NCT04886596, NCT05035212, NCT05127434, NCT03982199, NCT02508194) on five RSV vaccines (RSVPreF3, RSVpreF, mRNA-1345, Ad26.RSV.preF, MEDI7510; see Table A3). Of the retrieved studies, six reported on the first season after the delivery of the RSV vaccine [51,52,80,81,82,83], two studies reported on follow-up studies [51,61], and one study reported on both primary and follow-up studies [48]. Overall, seven studies were published as peer-reviewed reports [48,51,52,80,81,82,83], while two studies were reported as conference abstracts [51,61]. A summary of the retrieved studies is provided in Table 1.

As the studies of Ison et al. [48], Feldman et al. [80], and Papi et al. [81] for the first RSV season insisted on using the very same population, pooled estimates were calculated from the study of Ison et al. [48]. Finally, the studies involved a total of 101,931 subjects older than 60 years (51,055 vaccinated vs. 50,876 placebo) for the first season, and 33,130 vaccinated vs. 38,068 placebo individuals for the follow-up. In this regard, while the study from Ison et al. [48] (RSVPreF3) reported on the full second season, to date only mid-season 2 data have been published for RCT NCT05035212 [61] (RSVpreF) and estimates for 12 months after the delivery of the first dose for NCT04886596 [85].

Assessed outcomes are summarized in Table 2. Briefly, four out of five RCT studies and seven out of nine studies only included RT-qPCR confirmed RSV cases [48,51,80,81,82,85,86], while in the RCT on RSVpreF, RSV infection was confirmed through either RT-qPCR or NAAT [52,61].

Even though all studies reported on the cases of symptomatic ARI, the main outcome of the included RCTs was the first episode of RSV-associated LRTD. While three RCTs (NCT04886596, NCT05035212, and NCT03982199) reported on LRTD defined by either two or more than three signs/symptoms [51,61,82,85,87], and NCT02508194 [83] reported on cases of LRTD with two or more signs/symptoms and on cases of ARI with one symptom or more from two or three different locations, RCT on RSVPreF3 (NCT04886596) implemented a different case definition (see Table A4). To begin with, researchers strictly dichotomized RSV-associated findings in signs and symptoms. Then, an LRTD was defined by either one sign and one symptom, or by a combination of three signs and/or symptoms, but discrete data were not provided. For the aim of the present review, LRTD cases from NCT04886596 were therefore pooled together with LRTD cases with three signs and/or symptoms rather than with cases with two signs/symptoms.

### 3.2. Risk of Bias

The overall quality of the included studies is detailed in Table 1 and summarized in Figure A1. Overall, all studies were characterized by high quality and low risk of bias. A partial exception is represented by the studies from the RCT NCT04886596 [48,80,81] because of the inconsistencies in the case definition when compared to other cases and the potential inclusion of more severe cases (i.e., those with three or more signs/symptoms) with milder ones, as well as by the limited information conveyed by conference reports [61,85].

### 3.3. Demographic Characteristics of the Included Studies

As shown in Table 3, there were no statistically significant differences between vaccine recipients and subjects treated with placebo in terms of prevalence of white ethnicity (74.50% in vaccinated, 74.64% in placebo, RR 0.996, 95% CI 0.990 to 1.003; I 0.0%, 95% CI 0.0 to 79.2; Q 2.09, *p* = 0.300) and subjects older than 70 years (or 75 years according to the age groups reported; 37.84% vs. 37.77%, RR 1.000, 95% CI 0.984 to 1.015; I 0.0%, 95% CI 0.0 to 79.2; Q 0.18, *p* = 0.996). On the contrary, while no substantial difference in terms of pooled prevalence of male gender was identified among the included studies (49.72% vs. 49.25%, RR 1.024, 95% CI 0.987 to 1.061), these estimates were affected by substantial heterogeneity (71.2%, 95% CI 27.1 to 88.6; Q 13.89, *p* = 0.008). As summarized in Table A5, the proportion of males ranged from 41.90% (vaccine group) vs. 34.45% (placebo) in the study by Falloon et al. [83] to 51.12% vs. 50.39% (for the vaccine and placebo groups, respectively) in the study by Walsh et al. [52].

Moreover, a detailed report on comorbidities was provided by four RCTs for COPD and congestive heart disease [48,51,52,82], and for asthma by three studies [48,52,82], while the remaining studies more properly included a description of “respiratory” and “cardiovascular” disorders (Table A6). Again, no substantial differences were identified.

### 3.4. Effectiveness of RSV Vaccine

#### 3.4.1. First Season

The effectiveness of the assessed vaccines is summarized in Table 4, while individual estimates are reported in forest plots included as Figure A2, Figure A3, Figure A4, Figure A5, Figure A6, Figure A7, Figure A8, Figure A9, Figure A10, Figure A11, Figure A12, Figure A13, Figure A14, Figure A15 and Figure A16. Regarding LRTD, distinctive estimates were calculated for LRTD with two symptoms and with three or more symptoms (see Table A4 for details).

##### VE on ARI

The reported VE for preventing ARI ranged from −5.08% (95% CI −106.67 to 46.57) of MEDI7510 to 69.66% (95% CI 43.70 to 83.65), with a pooled estimate of 59.88% (95% CI 41.17 to 72.64), which was affected by substantial heterogeneity (I^2^ 61.0%, 95% CI 0.0 to 85.2). Estimates for the VE on RSV A and B were provided for RSVpreF, mRNA-1345, and MEDI7510 [51,52,83] (Figure A3 and Figure A4): Individual efficacy for RSV A was 78.50% (95% CI 58.77 to 88.79), 66.95% (95% CI −2.46 to 89.34), and −5.08% (95% CI −106.67 to 46.57) for mRNA-1345, RSVpreF, and MEDI7510, respectively, with a pooled estimate of 57.17% (95% CI −16.28 to 84.23). Individual VE for RSV B was 51.77% (95% CI 10.69 to 73.95), 82.37% (95% CI 62.62 to 91.69), and −34.86% (95% CI −50.11 to 84.37) for mRNA-1345, RSVpreF, and MEDI7510, respectively, with a pooled estimate of 51.56% (95% CI −50.11 to 84.37). Both estimates were affected by substantial heterogeneity (I^2^ 82.2%, 95% CI 45.3 to 94.2, and 85.4%, 95% CI 57.1 to 95.0 for RSV A and RSV B, respectively). Calculation of RR for ARI associated with RSV A vs. RSV B ruled out significant differences in the corresponding VE (RR 0.849, 95% CI 0.529 to 1.363; I^2^ 0.0%, 95% CI 0.0 to 89.6, Q = 1.56, *p* = 0.458) (Table 5). 

When focusing on adults older than 70/75 years at the time of recruitment [51,52,82], individual estimates were available for RSVpreF, mRNA-1345, and Ad26.RSV.preF. As shown in Figure A6, VE ranged from 60.3% (95% CI −25.99 to 87.50) for Ad26.RSV.preF, to 62.31% (95% CI 14.98 to 83.29) for RSVpreF, and peaked at 84.23% (95% CI 62.71 to 93.33) for mRNA-1345, with a pooled estimate (72.31%, 95% CI 45.58 to 85.09) and a heterogeneity of 22.1% (95% CI 0.0 to 91.9).

##### VE on LRTD with 2 Symptoms

Point estimates (Figure A7) ranged from −34.86% (95% CI −192.00 to 37.72) for MEDI7510, to 66.95% (95% CI 34.63 to 83.29) for RSVpreF, to 74.91% (95% CI 49.93 to 87.43) for Ad26.RSV.preF, and peaked at 83.67% (95% CI 67.01 to 91.94) for mRNA-1345. A pooled estimate of 63.66% (95% 12.35 to 84.93) was therefore calculated, with substantial heterogeneity (I^2^ 82.5%, 95% CI 55.0 to 93.2, *p* < 0.001). A subanalysis for individuals aged >70/75 years at the time of the recruitment (see Figure A8) was provided from three studies [51,52,82], and identified a VE ranging from 78.80% (95% CI 26.27 to 93.91 for RSVpreF) to 80.16 (95% CI 9.74 to 95.64) for Ad26.RSV.preF and 95.46% (95% CI 66.32 to 99.39) for mRNA-1345. The corresponding pooled estimate was 84.46% (95% CI 63.00 to 93.47). 

##### VE on LRTD with 3 Symptoms or More

Point estimates (Figure A9) were provided for Ad26.RSV.preF (79.93%, 95% CI 51.85 to 91.63), RSVPreF3 (80.91%, 95% CI 64.62 to 89.70), and mRNA-1345 (82.41%, 95% CI 39.99 to 94.84), peaking at 85.84% (37.69 to 96.78) for RSVpreF, with a pooled VE of 81.38% (95% CI 70.94 to 88.06). Even though the point estimate suggested a low heterogeneity (I^2^ = 0.0%), a contradictory report was provided by 95% CI (0.0 to 84.7). The pooled efficacy against RSV A was 83.76% (95% CI 52.95 to 94.40; Figure A10), while the pooled VE for RSV B was estimated as 80.72% (95% CI 58.79 to 90.98; Figure A11), with no substantial differences, as shown in Table 5 (RR 0.505; 95% CI 0.150 to 1.706 for LRTD with three or more symptoms in RSV A vs. RSV B; Figure A12).

Finally, subanalysis for individuals aged 70/75 years or more identified a pooled VE of 83.78% (95% CI 61.43 to 93.18) that resulted from four point estimates, i.e., RSVPreF3 (84.18%, 95% CI 46.95 to 95.32) [81], mRNA-1345 (92.32%, 95% CI −36.39 to 99.57) [51], RSVpreF (91.01%, 95% CI −62.62 to 99.50) [52], and Ad26.RSV.preF (75.20%, 95% CI −16.42 to 94.72) [82].

##### Comparison of Main Outcome by Age Groups

As shown in Table 6, the risk for all assessed outcomes was similar in subjects aged 70/75 years or more and among younger subjects (i.e., <70/75 year of age: RR 0.800, 95% CI 0.459 to 1.392; RR 0.522, 95% CI 0.211 to 1.291; RR 0.819, 95% CI 0.304 to 2.207 for ARI, LRTD with two symptoms, and LRTD with three or more symptoms, respectively) (Figure A15 and Figure A16).

#### 3.4.2. Follow-Up Studies

The three studies providing data on the follow-up of vaccinated individuals [48,61,85] provided pooled estimates for VE regarding the occurrence of ARI and LRTD with three or more symptoms (see Table A4 for case definition). More precisely, the VE on ARI was estimated as 46.64% (95% CI 35.94 to 55.55) and as 61.15% (45.29 to 72.40) for LRTD with three or more symptoms. Individual estimates for ARI ranged from 40.04% (95% CI 18.90 to 55.67) for RSVPreF3 to 41.38% (95% CI 16.97 and 58.62) for RSVpreF, and peaked at 53.69% (95% CI 40.24 to 64.11) for mRNA-1345, while estimates for LRTD with three or more symptoms was 55.83% (95% CI 28.41 to 72.75), 78.65% (95% CI 25.72 to 93.86), and 62.88% (95% CI 37.17 to 78.07) for RSVpreF3, RSVpreF, and mRNA-1345, respectively (Figure A17 and Figure A18). Heterogeneity among the retrieved studies was reportedly low in point estimates, while the 95% CI for I^2^ exceeded the cutoff for substantial heterogeneity (0.0 to 89.9 and 0.0 to 89.6, respectively). Subanalyses were performed on the efficacy for the prevention of LRTD with three symptoms or more in the RSV serogroup (Figure A19 and Figure A20), and the corresponding pooled estimates for VE were 70.55%, 95% CI 45.51 to 84.08) for RSV A compared to 50.25% (95% CI 22.62 to 68.02) for RSV B. Not coincidentally (Figure A21), the point estimates for RSVpreF and mRNA-1345 were significantly similar for RSV A and RSV B (RR 1.000, 95% CI 0.063 to 15.985).

The risk for breakthrough infections during the follow-up period is summarized in Table 7 and detailed in Figure A22 and Figure A23. As shown, the risk substantially increased for both ARI (RR 3.740, 95% CI 2.875 to 4.866) and LRTD with three or more symptoms (RR 4.326, 95% CI 2.415 to 7.748). Even though heterogeneity was not substantial when taken into account as a point estimate (I2 = 0.0% for both ARI and LRTD with three or more symptoms), a contradictory appraisal was suggested by the corresponding 95% CI that in both cases exceeded the cutoff value of 60%. When focusing on the individual point estimates, no significant differences were identified for the primary season vs. follow-up season for RSVpreF (RR 2.575, 95% CI 0.430 to 15.410).

Data on cumulative estimates were similarly provided. However, as data on RSVpreF only included 6 months of season two (representing “mid-season” estimates), and the RCT on mRNA-1345 eventually reported on a total of 12 months across two RSV seasons, a pooled meta-analysis was not performed. However, cumulative estimates across two seasons (Figure 2) were then calculated as 67.73% (59.34 to 74.39), 52.42% (95% CI 38.64 to 63.11), and 53.59% (95% CI 40.23 to 64.11) for RSVPreF3, RSVpreF, and mRNA-1345, respectively, on the prevention of ARI. On the other hand, corresponding estimates for the prevention of LRTD with three or more symptoms were 78.38% (95% CI 67.93 to 85.41), 84.36% (59.88 to 93.91), and 62.88% (37.17 to 78.07) for RSVPreF3, RSVpreF, and mRNA-1345, respectively.

### 3.5. Sensitivity Analysis

Sensitivity analysis required the removal of single studies at a time, and resulting pooled estimates are reported in Figure A24, Figure A25, Figure A26, Figure A27, Figure A28, Figure A29, Figure A30, Figure A31, Figure A32, Figure A33, Figure A34, Figure A35 and Figure A36. Where included in the pooled analyses, the removal of the study by Falloon et al. [83] on MEDI7510 led to a substantial increase in VE (ARI: 66.43%, 95% CI 57.55 to 73.45; LRTD with two symptoms 79.99%, 95% CI 64.18 to 83.91), and heterogeneity decreased consistently (0.0% for ARI, 1% for LRTD with two symptoms). Contrarily, the removal of the study by Wilson et al. on mRNA-1345 [85] from follow-up pooled estimates led to a reduction in VE (40.62%, 95% CI 25.40 to 52.73) for ARI, while the removal of the study by Walsh et al. on RSVpreF [61] reduced the VE to 59.20% (95% CI 41.77 to 71.42).

### 3.6. Publication Bias

Publication bias was initially ascertained by calculation and visual inspection of funnel plots. In funnel plots, the sample size is plotted against the effect size they report. As the size of the sample increases, individual estimates of the effect are likely to converge around the true underlying estimate [63,66,73]. Funnel plots for the primary season (for ARI and LRTD with two and three or more symptoms) are reported in Figure 3.

All funnel plots were substantially asymmetrical, as all the points did point towards the left half of the plot, suggesting the presence of publication bias with a high share of lower prevision studies. However, Egger’s test (Table 8) substantially ruled out potential publication bias, suggesting that other potential sources of asymmetry (e.g., heterogeneity due to different choices in the outcome measure, differences in the underlying risk, etc.) should be ascertained. 

Despite the reduced number of included studies, radial plots were seemingly spared by the clustering of retrieved data, with individual estimates scattered across both sides of the regression lines (Figure 4).

Regarding follow-up studies (Figure 5), the analysis of funnel plots for ARI (Figure 5a) and LRTD with three or more symptoms (Figure 5c) suggested a similar asymmetry of estimates, but Egger’s test substantially ruled out the potential publication bias as an explanation of these findings. Also, radial plots (Figure 5b,d) were somehow scattered across the regression line, but the limited number of observations suggested a cautious appraisal of visual inspection. 

## 4. Discussion

### 4.1. Summary of Main Findings and Their Generalizability

In this systematic review with meta-analysis, we retrieved and ultimately included a total of nine studies on five RSV vaccines (RSVpreF, RSVPreF3, MEDI7510, Ad.26.RSV.preF, and mRNA-1345). Of those, four formulates were based on prefusion F protein (RSVpreF, RSVPreF3, Ad.26.RSV.preF, and mRNA-1345). The only post-fusion F protein-based vaccine to date has been officially discontinued (MEDI7510) [83,86,89]. 

All the full papers included in this review were of high or even very high quality, and the risk of literature bias was substantially low. Collected data were meta-analyzed, focusing on the VE for the prevention of LRTD in older adults compared with that in the placebo groups, and sub-analyses were specifically performed on adults aged more than 70 or 75 years, according to the age groups included in the primary studies. Heterogenous definitions of LRTD were used in the included studies, particularly when dealing with symptoms that were included in the respective definition (see Table A4). Taking into account this potential source of heterogeneity, a pooled estimate for VE of 81.38% (95% CI 70.94 to 88.06) for LRTD with three or more symptoms was eventually calculated for mRNA-1345, RSVpreF, RSVPreF3, and Ad.26.RSV.preF, with no substantial difference for RSV A (83.76%, 95% CI 52.95 to 94.40) or RSV B (80.72%, 95% CI 58.79 to 90.98), as the RR for LRTD with three or more symptoms associated with RSV A rather than RSV B was equal to 0.505 (95% CI 0.150; 1.706). 

Interestingly enough, the final efficacy in adults aged 70/75 years or older was estimated as 83.78% (95% CI 61.43 to 93.18), with no substantial differences in younger individuals (RR 0.522, 95% CI 0.211 to 1.291). In other words, the four assessed formulates were not only quite effective in preventing more severe cases of RSV-associated diseases in older adults, but sensitivity analysis also suggests similar VE estimates.

The assessed RSV vaccines appeared to be significantly less effective against ARI than against LRTD, as the pooled efficacy was estimated in 59.88% (95% CI 41.17 to 72.64), which was improved to 66.43% (95% CI 57.55 to 73.45) after the removal of the report on MEDI7510. In this regard, it should be stressed that the WHO preferred product characteristics for RSV vaccines [38,90] were initially designed for infants and newborns, and a 50% VE in the prevention of severe RSV disease due to LRTD was considered an acceptable target. 

Follow-up data were available for RSVpreF, RSVPreF3, and mRNA-1345, including a full report on the second RSV season after the delivery of the vaccine for RSVPreF3 [48], a mid-term report on the second RSV season for RSVpreF [61], and a report on the first calendar year after the delivery of the RSV vaccine for mRNA-1345 [85]. However, despite the promising results from the first RSV season [82,91], no updates on the efficacy of Ad26.RSV.preF vaccine have been published since 2023 [82,92,93]. As the Ad26.RSV.preF was designed as an adenovirus vector-based vaccine, we cannot rule out that it may have been affected by the “fallout” that followed COVID-19 vaccination campaigns. The vaxzevria vaccine for SARS-CoV-2 is similarly based on replication-defective adenovirus (ChAdOx1), and claims for increased risk for thromboembolic events in patients having received this immunization [94] have possibly forced the strengthening of its preventive assessment [82,92,93,95,96]. Moreover, follow-up data on RSVpreF and mRNA-1345 have only been published as conference abstracts [61,85]. Despite their potential significance from a public health point of view, the content they conveyed should be quite cautiously appraised, as conference proceedings should be acknowledged as a “primer” for more detailed primary research, thereby emphasizing the need for more studies. This may explain the lack of information in publications that must be acknowledged as preliminary reports. Taking into account all of the aforementioned potential shortcomings, the pooled VE for ARI was 46.64% (35.95 to 55.55), compared to 61.15% (45.29 to 72.40) for LRTD with three or more symptoms [48,61,85]. In other words, the risk for breakthrough infections was significantly higher after the primary season not only for ARI (RR 3.740, 95% CI 2.875 to 4.866) but also for LRTD (RR 4.326, 95% CI 2.415 to 7.748). When data were considered across the whole of the assessed time period, mRNA-1345 was associated with a cumulative VE of 53.59% (40.23 to 64.11) for ARI and 62.88% (37.17 to 78.07) for LRTD with three or more symptoms compared to 67.73% (59.34 to 74.39) for ARI and 78.38% (67.93 to 85.41) for LRTD with three or more symptoms for RSVPreF3 and 52.42% (38.64 to 63.11) and 84.36% (59.88 to 93.91) for ARI and LRTD with three or more symptoms among subjects vaccinated with RSVpreF.

In fact, the reported VE was quite heterogeneous, as suggested by the analysis of 95% CI for I^2^ statistics, and several explanations could be suggested. Most notably, the observation time for RSVPreF3 included two full RSV seasons [48,80], while available data for RSVpreF were limited to the end of RSV season in the USA, representing a mid-season term [61], and estimates for mRNA-1345 represented an additional analysis collected 12 months after the vaccine, irrespective of the underlying seasonal epidemiology of RSV [85]. Nonetheless, a provisionary estimate for two end-season LRTD cases among RSVpreF vaccine recipients has recently been reported, including a preliminary VE equal to 77.8% (95.0% CI: 51.4, 91.1) [88]. Unfortunately, as the complete data are still not available, not only were such estimates not included in our systematic review and meta-analysis, but also, due the high heterogeneity in the overall length of the assessment, a pooled analysis for the across-season efficacy was not performed.

Another explanation is associated with the uneven sample size. In the study from Ison et al. [48], nearly half of the original vaccine group was randomized to receive a second dose of RSV vaccine and was therefore removed from the corresponding estimates, which were based on only 4991 vaccinated individuals (i.e., 40.02% of the original sample) and 10,031 placebos (i.e., 80.06% of the original sample). On the contrary, the RCT on the RSVpreF vaccine retained 58.24% and 58.54% of the vaccine and placebo groups, respectively, each one encompassing around 10,000 subjects, while the RCT on mRNA-1345 included more than 36,000 individuals, which is nearly the same number of cases as in the other two clinical trials combined. Finally, the overall epidemiology of RSV should be accurately considered. For example, the ratio of RSV-A vs. RSV-B-associated ARI in the placebo group was 0.267 in the first season of RCT on RSVpreF and 2.480 in mid-season two. On the contrary, the RSV A/B ratio in mRNA-1345 was the highest for the first series of cases, corresponding to the first RSV season, and the lowest in the supplementary analysis (1.645 vs. 1.325) (Table A7). As RSV cases reported from placebo groups were not influenced by characteristics of RSV vaccines, the aforementioned heterogeneities could be explained by underlying features of RSV epidemiology in the countries where the RCTs were performed.

### 4.2. Limits and Implications for Future Studies

Even though our study provides a pooled estimate of VE of the available formulates against RSV, being of potential significance for both public health and healthcare professionals, we acknowledge a number of limitations.

Firstly, we applied a strict search strategy on a relatively new topic (i.e., RSV vaccines) that guaranteed the good or very good quality of retrieved studies but resulted in a small number of collected studies. Notably, the very same shortcoming was identified in a similarly designed systematic review with meta-analysis focusing on maternal vaccination [65].

Second, even though most included studies were of appropriate or even of high quality, we gathered data on diverse vaccine strategies, and the variable vaccine type investigated in the retrieved RCTs reasonably introduced high heterogeneity in the results, potentially affecting the final results. Not coincidentally, estimates of the immunogenicity of the assessed vaccines were irregularly provided across the retrieved studies. A summary is provided in Table A8. In this regard, despite the differences in vaccine strategy and heterogeneity in the reporting strategies, all included vaccines were documented as highly immunogenic, at least during the first months after the delivery of the vaccine.

Thirdly, the collected studies were consistently heterogenous not only regarding the inquired vaccine but also in terms of timeframe, geographical settings, sample size, and reporting strategy. This specific topic should be accurately stressed. On the one hand, as summarized in Table A3 and Table A4, for some vaccines (i.e., RSVpreF, mRNA-1345, and Ad26.RSV.preF), VE was calculated regarding the incident ARI and LRTD with two and three or more symptoms [51,52,87,91], while for others (RSVPreF3) a case definition was employed as presenting two mild symptoms plus at least one severe symptom [48,80,81]. On the other hand, the very same nature of the included symptoms was in turn inconsistent across studies. For example, the NCT05035212 (also known as the RENOIR clinical trial) included a total of five symptoms (i.e., cough, wheezing, sputum, shortness of breath, tachypnea), two of which (i.e., cough and sputum) are usually associated with upper respiratory tract infections and are usually the first to appear during the RSV clinical syndrome, irrespective of its eventual severity [5,97,98,99,100]. Other studies (i.e., NCT03982199 [82], NCT02508194 [83], and NCT04886596 [48,80,81]) not only relied on a larger panel of signs and symptoms but also included in the case definition a series of systemic or lower respiratory tract findings. As a consequence, when dealing with LRTD defined by only two findings from a broader range of signs and symptoms, it is more likely to include only upper respiratory symptoms like sputum or cough, while a case definition with three or more findings may be associated with a more severe syndrome, as it would reasonably include signs and symptoms such tachypnea/breath shortness or wheezing. Not coincidentally, the Advisory Committee on Immunization Practices (ACIP) not only recommended both licensed RSV vaccines (RSVpreF and RSVPreF3) [43] but also considered LRTD with two symptoms from NCT04886596 [48,80,81] and LRTD with three or more symptoms from NCT05035212 [52] to be equal endpoints in their rationale. 

Fourthly, as stressed by Melgar et al. [43], all available studies were deprived of sufficient statistical power to ascertain the efficacy of the included vaccines in averting RSV-related hospitalizations and deaths in older adults, particularly among adults aged 80 years or older. As previously outlined [20,101,102,103], RSV infections in older adults can result in quite severe and potentially lethal complications, particularly among individuals affected by cardiorespiratory comorbidities [43,104,105]. For example, in their report on the disease severity of RSV compared with COVID-19 and influenza, adults older than 60 years hospitalized because of RSV were more likely to require oxygen therapy, non-invasive ventilation, or admission to the intensive care unit than those hospitalized due to COVID-19 (adjusted odds ratio (aOR) 2.97 95% CI 2.07–4.27, aOR 2.25 95% CI 1.65–3.07, and aOR 1.49, 95% CI 1.13–1.97, respectively) or influenza (aOR 2.07 95% CI 1.37–3.11, aOR 1.99, 95% CI 1.36–2.90, and aOR 1.55, 95% CI 1.11–2.19, respectively) [106]. Similarly, a study from 25 hospitals in France on a total of 1168 adults and elderly individuals hospitalized for RSV infections from 1 January 2015 to 31 December 2019 reported a 25% rate of ICU admissions [107]. In this regard, another limit of the present study is related to the reported prevalence of comorbidities included in parent studies. In fact, there is considerable evidence that obesity, hypertension, chronic heart failure, COPD, and chronic respiratory failure are associated with an increased risk of ICU admission due to RSV infections [107]. Still, data on congestive heart disease, COPD, and asthma from the retrieved studies [48,51,52,82] hinted at a relatively lower rate of these comorbidities (6.65%, 1.60%, and 9.21%, respectively) compared to the general populations of the European Union and United States (i.e., the economic areas where RSV have been initially licensed and may be considered particularly suitable because of the high prevalence of older adults), particularly for COPD. For example, in 2019, an estimated worldwide COPD mean prevalence was identified as 13.1% (10.2–15.6%), with the following distribution by continents: Europe, 12.4% (8.8–16.0%); Africa, 13.9% (12.0–15.9%); America, 13.2% (10.5–15.9%); Asia, 13.5% (10.0–16.0%); and Oceania, 11.6% (9.8–13.1%) [108].

Fifthly, even though adults aged 80 years or older are likely to benefit the most from RSV vaccines [20,28,104], this age group was substantially under-represented in all of the assessed studies. Therefore, the potential role of RSV vaccines from a public health point of view should be carefully evaluated, particularly when tailoring potential strategies for age groups at higher risk for RSV complications [43].

Sixthly, our data should be reconciled with clinical features and diagnostic options of RSV infection. On the one hand, none of the included RCTs performed an active case search for incident infections but rather targeted symptomatic ones. On the other hand, even though RT-qPCR has been often considered the “gold standard” for the diagnosis of viral respiratory infections [109,110,111], the strategy for the collection of testing specimens could fail to achieve an appropriate diagnosis in accordance with the stage of the assessed infection [112,113]. More precisely, in severe cases associated with lower respiratory tract infections, RT-qPCR testing based on nasal swabs may result in false negative results, as the viral infection is more likely to be active in the lower regions of the respiratory tract [113]. In these cases, the addiction of sputum, paired serology, and oropharyngeal swabs would radically improve the diagnostic efficacy (+52%, + 44%, and +28%, respectively) [113], but none of the included RCTs actually implemented the aforementioned options. In other words, we cannot rule out that point estimates for VE, and the resulting pooling VE estimates, would been inflated by false negative RSV cases.

Finally, when addressing the potential use of RSV vaccines in real-world settings, not only VE but also the safety profile of interventions should be considered. In our study, we deliberately focused on VE. Even though all RCTs included in the study hinted at an acceptable safety profile [48,51,52,80,81,83,87,89,91], early reports from maternal vaccination [114,115,116] highlighted an increased risk for preterm birth, suggesting a previously unexpected health impact. The vaccines included in maternal studies were RSVpreF and RSVPreF3 in the very same formulate for adult vaccination, and therefore a continued focus on balancing the risks and potential benefits will be essential in tailoring future vaccination strategies. 

## 5. Conclusions

RSV vaccination of older adults has been proven to be effective in preventing LRTD cases, particularly when considering more severe cases with three or more signs or symptoms. However, the limited statistical power of the included studies on hospitalizations and severe complications, including deaths, the lack of complete and fully comparable follow-up seasons for all of the retrieved studies, and the failure to adopt a uniform and appropriate case definition across studies, altogether stress the importance of future and improved reporting from ongoing RCTs.

## Figures and Tables

**Figure 1 vaccines-12-00500-f001:**
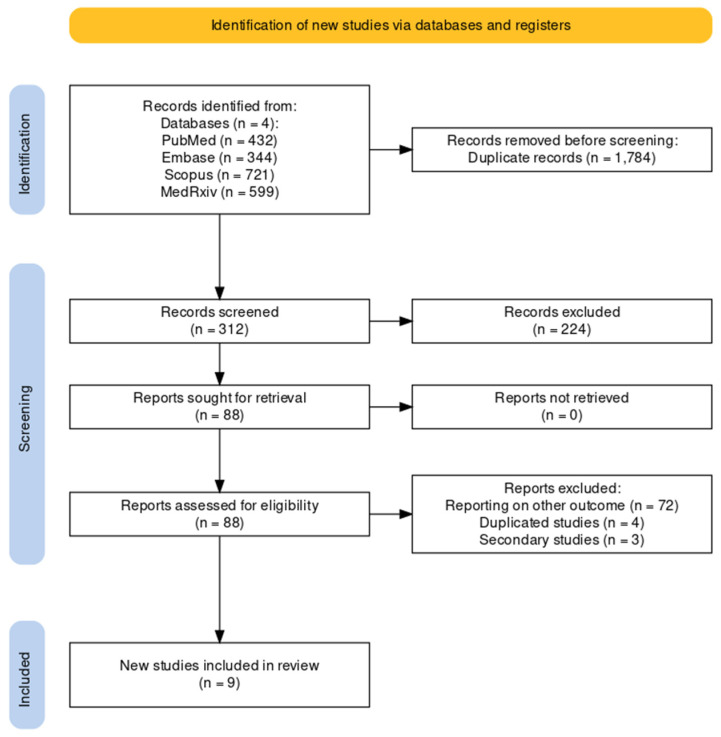
Flowchart of included studies.

**Figure 2 vaccines-12-00500-f002:**
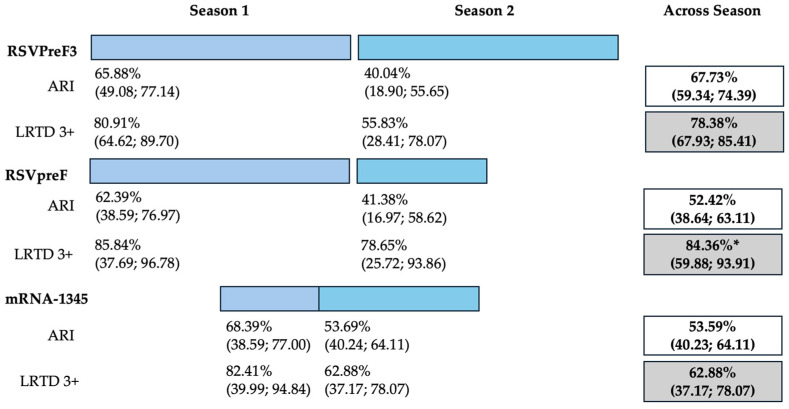
Follow-up studies and vaccine efficacy estimates for season 1, season 2, and across seasons on the prevention of acute respiratory infection (ARI) and lower respiratory tract disease (LRTD) with 3 or more signs/symptoms. All results are reported as point estimates in percent value and corresponding 95% confidence intervals. * = provisionary estimate for end of season 2 equals 77.8% (95.0% CI: 51.4, 91.1) [88].

**Figure 3 vaccines-12-00500-f003:**
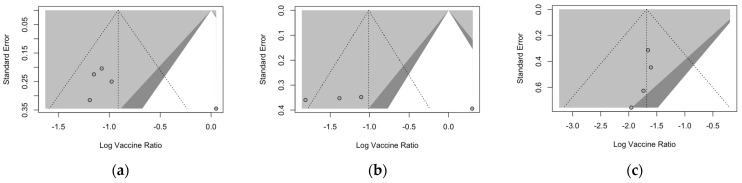
Funnel plots for randomized controlled trials included in the analyses. (**a**), acute respiratory illness; (**b**), lower respiratory tract disease with two symptoms; (**c**), lower respiratory tract diseases with three or more symptoms.

**Figure 4 vaccines-12-00500-f004:**
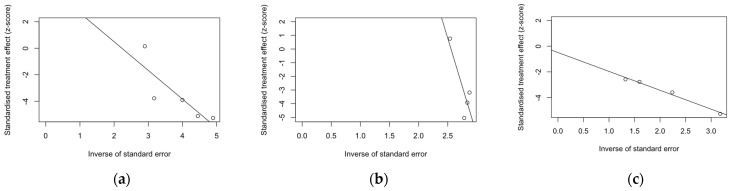
Radial plots for randomized controlled trials included in the analyses. (**a**), acute respiratory illness; (**b**), lower respiratory tract disease with two symptoms; (**c**), lower respiratory tract disease with three or more symptoms.

**Figure 5 vaccines-12-00500-f005:**
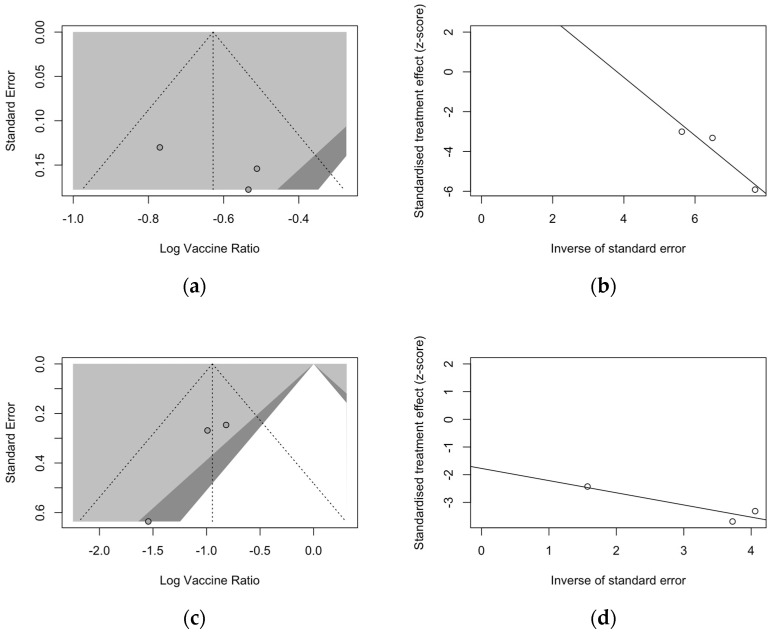
Funnel and radial plots and for randomized controlled trials included in the analyses and dealing with follow-up studies for ARI (**a**,**b**) and for lower respiratory tract diseases with three or more symptoms (**c**,**d**).

**Table 1 vaccines-12-00500-t001:** Summary of randomized controlled trials (RCTs) on vaccines against respiratory syncytial virus (RSV), including the summary of the risk of bias assessment according to the risk of bias (ROB) tool from the National Toxicology Program (NTP)’s Office of Health Assessment and Translation (OHAT) handbook [74,84] on observational studies included in the meta-analysis. Details on individual characteristics of the vaccines are provided in Table A3. Note: D1: possibility of selection bias; D2: exposure assessment; D3: outcome assessment; D4: confounding factors; D5: reporting bias; D6: other bias; PreF = prefusion, PostF = post-fusion; Adj = adjuvated; FP = full paper; AR = abstract report.

Study	Vaccine	Study Period(from … to)	Countries	RCT	Season	Vaccinated(N.)	Placebo(N.)	ROB Assessment
D1	D2	D3	D4	D5	D6
Studies reporting on the first season after the delivery of RSV vaccine
Ison et al. [48]FP	RSVPreF3	Subunit, PreF, Adj, MV (A)	25/05/202130/09/2023	17	Phase 3NCT04886596	1 + 2	12,470	12,503			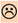		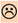	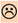
Feldman et al. [80]FP	RSVPreF3	Subunit, PreF, Adj, MV (A)	25/05/202131/01/2022	17	Phase 3NCT04886596	1	12,466	12,494			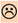			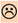
Papi et al. [81]FP	RSVPreF3	Subunit, PreF, Adj, MV (A)	25/05/202131/01/2022	17	Phase 3NCT04886596	1	12,467	12,499			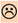			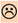
Walsh et al. [52]FP	RSVpreF	Subunit, PreF, BV (A + B)	31/08/202114/07/2022	7	Phase 3NCT05035212	1	17,215	17,069						
Wilson et al. [51]FP	mRNA-1345	mRNA;PreF; MV (A)	17/11/202131/10/2022	22	Phase 2/3NCT05127434	1	17,734	17,679					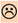	
Falsey et al. [82]FP	Ad26.RSV.preF	Vector-basedPreF; MV (A)	05/08/201920/03/2020	1	Phase 2bNCT03982199	1	2891	2891						
Falloon et al. [83]FP	MEDI7510	Protein-basedPostF; MV (A)	30/09/201509/09/2016	7	Phase 2NCT02508194	1	949	951						
Follow-up studies
Ison et al. [48]FP	RSVPreF3	Subunit, PreF, Adj, MV (A)	25/05/202130/09/2023	17	Phase 3NCT04886596	Full season 2	4991	10,031			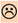		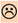	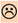
Walsh et al. [61]AR	RSVpreF	Subunit, PreF, BV (A + B)	31/08/2021undefined	7	Phase 3NCT05035212	Mid-season 2	10,027	9992						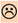
Wilson et al. [85]AR	mRNA-1345	mRNA;PreF; MV (A)	17/11/2021undefined	22	Phase 3NCT04886596	12 months afterdelivery	18,112	18,045					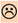	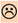


 = definitively low; 

 = probably low; 
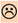
 = probably high.

**Table 2 vaccines-12-00500-t002:** Summary of outcome definition for lower respiratory tract disease (LRTD). Note: RT-qPCR = real-time quantitative polymerase chain reaction; NAAT = nucleic acid amplification test (i.e., isothermal amplification tests). Likewise: nicking endonuclease amplification reaction, NEAR; transcription mediated amplification, TMA; loop-mediated isothermal amplification, LAMP; helicase-dependent amplification, HAD; clustered regularly interspaced short palindromic repeats, CRISPR; strand displacement amplification, SDA.

Trial (Vaccine)	RSV Confirmation	Endpoint	Characteristics
NCT03982199 (Ad26.RSV.preF)	RT-qPCR	1	VE against first episode of LRTD defined as 3 or more symptoms of LRTD
		2	VE against first episode of LRTD defined as 2 or more symptoms of LRTD
		3	VE against first episode of LRTD defined as 1 or more symptoms of LRTD + systemic symptoms
NCT02508194 (MEDI7510)	RT-qPCR	1	VE against first episode of ARI plus ≥ 1 symptoms from any 2 or 3 locations
		2	VE against first episode of LRTD defined as 2 or more symptoms of LRTD
NCT04886596 (RSVPreF3)	RT-qPCR		VE against first episode of LRTD defined as at least 1 sign + 1 further sign or symptom of LRTD OR VE against first episode of LRTD defined as at least 3 signs OR symptoms of LRTD
NCT05035212 (RSVpreF)	RT-qPCR OR NAAT	1	VE against first episode of LRTD defined as at least 2 signs/symptoms of LRTD
		2	VE against first episode of LRTD defined as at least 3 signs/symptoms of LRTD
NCT04886596 (mRNA-1345)	RT-qPCR	1	VE against first episode of LRTD defined as at least 2 signs/symptoms of LRTD
		2	VE against first episode of LRTD defined as at least 3 signs/symptoms of LRTD

**Table 3 vaccines-12-00500-t003:** Demographics and clinical characteristics of the subjects included in the primary studies (only cases in the first season).

Variable	Studies	No. of Subjects (/Total, %)	No. of Placebo (/Total, %)	RR (95% CI)	Heterogeneity		
					I^2^ (95% CI)	τ^2^ (95% CI)	Q (*p*-Value)
Demographic data							
Age > 70/75 years	[48,51,52,80,81,82,83]	19,316/51,055 (37.83%)	19,216/50,876 (37.77%)	1000 (0.984; 1.015)	0.0% (0.0; 79.2)	0.001 (0.001; 0.002)	0.18 (0.996)
White ethnicity	[48,51,52,80,81,82,83]	38,034/51,055 (74.50%)	37,975/50,876 (74.64%)	0.996 (0.990; 1.003)	0.0% (0.0; 79.2)	0.001 (0.001; 0.002)	2.09 (0.300)
Male Gender	[48,51,52,80,81,82,83]	25,386/51,055 (49.72%)	25,056/50,876 (49.25%)	1.024 (0.987; 1.061)	71.2% (27.1; 88.6)	0.001 (0.000; 0.056)	13.89 (0.008)
Comorbidities							
COPD	[48,51,52,82]	3322/49,979 (6.65%)	3379/50,148 (6.74%)	0.986 (0.942; 1.033)	0.0% (0.0; 84.7)	0.001 (0.000; 0.024)	0.38 (0.945)
Congestive heart disease	[48,51,52,82]	802/49,979 (1.60%)	815/50,148 (1.63%)	0.986 (0.896; 1.086)	0.0% (0.0; 84.7)	0.001 (0.000; 0.015)	0.38 (0.945)
Asthma	[48,52,82]	3000/32,576 (9.21%)	2871/32,463 (8.84%)	1.049 (0.999; 1.101)	0.0% (0.0; 89.6)	0.001 (0.000; 0.014)	0.52 (0.772)

**Table 4 vaccines-12-00500-t004:** Summary of main outcomes of vaccine efficacy (VE) from collected studies (Note: 95% CI = 95% confidence interval; ARI = acute respiratory illness; LRTD = lower respiratory tract disease).

Outcome	Studies	No. of Subjects (/Total, %)	No. of Placebo (/Total, %)	VE (95% CI)	Heterogeneity		
					I^2^ (95% CI)	τ^2^ (95% CI)	Q (*p*-Value)
1st season, overall							
ARI	[48,51,52,82,83]	110/50,954 (0.22%)	293/50,781 (0.58%)	59.88% (41.17; 72.64)	61.0% (0.0; 85.2)	0.120 (0.000; 2.127)	10.15 (0.038)
LRTD (2 symptoms)	[51,52,82,83]	45/38,485 (0.12%)	139/38,283 (0.36%)	63.66% (12.35; 84.93)	82.5% (0.0; 93.2)	0.675 (0.122; 11.471)	17.14 (<0.001)
LRTD (3+ symptoms)	[48,51,52,82]	23/50,047 (0.05%)	124/49,884 (0.25%))	81.38% (70.94; 88.06)	0.0% (0.0; 84.7)	0.000 (0.001; 0.012)	0.17 (0.982)
1st season, RSV A							
ARI	[51,52,83]	32/35,694 (0.09%)	79/35,482 (0.22%)	51.17% (−16.28; 84.23)	82.2% (45.3; 94.2)	0.604 (0.062; 26.472)	11.25 (0.004)
LRTD (3+ symptoms)	[51,52,81]	4/47,253 (0.01%)	26/47,079 (0.06%)	83.76% (52.95; 94.40)	0.0% (0.0; 89.6)	0.000 (0.000; 13.311)	0.60 (0.741)
1st season, RSV B							
ARI	[51,52,83]	38/35,694 (0.11%)	87/35,482 (0.25%)	51.56% (−50.11; 85.37))	85.4% (57.1; 95.0)	0.866 (0.129; 40.735)	13.69 (0.001)
LRTD (3+ symptoms)	[51,52,81]	8/47,253 (0.02%)	43/47,079 (0.09%)	80.72% (58.79; 90.98)	0.0% (0.0; 89.6)	0.000 (0.000; 10.341)	0.64 (0.727)
1st season, only adults aged ≥70/75 years							
ARI	[51,52,82]	18/13,627 (0.13%)	69/13,546 (0.51%)	72.31% (48.58; 85.09)	22.1% (0.0; 91.9)	0.079 (0.000; 10.349)	2.57 (0.277)
LRTD (2 symptoms)	[51,52,82]	6/13,627 (0.04%)	46 13,546 (0.34%)	84.46% (63.00; 93.47)	0.0% (0.0; 89.6)	0.000 (0.000; 29.114)	1.79 (0.409)
LRTD (3+ symptoms)	[51,52,81,82]	5/19,130 (0.03%)	38/19,061 (0.20%)	83.78%) (61.43; 93.18)	0.0% (0.0; 84.7)	0.000 (0.000; 2.732)	0.71 (0.871)
Follow-up studies							
ARI	[48,61,85]	190/33,130 (0.57%)	451/38,068 (1.18%)	46.64% (35.94; 55.55)	1.8% (0.0; 89.9)	0.003 (0.000; 0.788)	2.04 (0.361)
LRTD (3+ symptoms)	[48,61,85]	42/33,130 (0.13%)	156/38,068 (0.41%)	61.15% (45.29; 72.40)	0.0% (0.0; 89.6)	0.003 (0.000; 5.411)	1.19 (0.553)
Follow-up studies, RSV A							
LRTD (3+ symptoms)	[48,61,85]	13/33,130 (0.03%)	53/38,068 (0.14%)	70.55% (45.51; 84.08)	0.0% (0.0; 89.6)	0.000 (0.000; 17.178)	1.18 (0.554)
Follow-up studies, RSV B							
LRTD (3+ symptoms)	[48,61,85]	26/33,130 (0.08%)	82/38,068 (0.22%)	50.25% (22.62; 68.02)	0.0% (0.0; 89.6)	0.000 (0.000; 2.270)	0.57 (0.750)

**Table 5 vaccines-12-00500-t005:** Summary of risk ratio (RR) for main outcomes from collected studies by serogroup of RSV infections (note: 95% CI = 95% confidence interval; ARI = acute respiratory illness; LRTD = lower respiratory tract disease).

Outcome	Studies	Total Samples	RSV A (/Total, %)	RSV B (/Total, %)	RR (95% CI)	Heterogeneity		
						I^2^ (95% CI)	τ^2^ (95% CI)	Q (*p*-Value)
1st season, RSV A vs. RSV B								
ARI	[51,52,83]	35,694	32 (0.09%)	38 (0.11%)	0.849 (0.529; 1.363)	0.0% (0.0; 89.6)	0.000 (0.000; 2.523)	1.56 (0.458)
LRTD (3+ symptoms)	[51,52,81]	47,253	4 (0.01%)	8 (0.2%)	0.505 (0.150; 1.706)	0.0% (0.0; 89.6)	0.000 (0.000; 7.469)	0.31 (0.856)
Follow-up studies, RSV A vs. RSV B								
LRTD (3+ symptoms)	[48,51,52]	33,130	13 (0.4%)	26 (0.8%)	0.499 (0.107; 2.327)	0.0% (0.0; 89.6)	1.167 (0.000; 59.266)	6.29 (0.043)

**Table 6 vaccines-12-00500-t006:** Summary of risk ratio (RR) for main outcomes from collected studies by age groups of participants (note: 95% CI = 95% confidence interval; ARI = acute respiratory illness; LRTD = lower respiratory tract disease).

Outcome	Studies	Subjects Aged ≥70/75 Years (/Total, %)	Subjects Aged <70/75 Years (/Total, %)	RR (95% CI)	Heterogeneity		
					I^2^ (95% CI)	τ^2^ (95% CI)	Q (*p*-Value)
First Season							
ARI	[51,52,82]	18/13,627 (0.13%)	43/23,951 (0.18%)	0.800 (0.459; 1.392)	0.0% (0.0; 89.6)	0.000 (0.000; 6.721)	1.40 (0.496)
LRTD (2 symptoms)	[51,52,82]	6/13,627 (0.04%)	24/23,951 (0.10%)	0.522 (0.211; 1.291)	0.0% (0.0; 89.6)	0.000 (0.000; 14.509)	0.84 (0.657)
LRTD (3+ symptoms)	[51,52,81,82]	5/19,130 (0.03%)	13/30,914 (0.04%)	0.819 (0.304; 2.207)	0.0% (0.0; 84.7)	0.000 (0.000; 7.504)	1.30 (0.729)

**Table 7 vaccines-12-00500-t007:** Summary of risk ratio (RR) for main outcomes from breakthrough infections by RSV in follow-up time period compared to primary season (note: 95% CI = 95% confidence interval; ARI = acute respiratory illness; LRTD = lower respiratory tract disease).

Outcome	Studies	Follow-Up (/Total, %)	Primary Season (/Total, %)	RR (95% CI)	Heterogeneity	Outcome	Studies
					I^2^ (95% CI)	τ^2^ (95% CI)	Q (*p*-Value)
ARI	[48,61,85]	190/33,130 (0.57%)	80/47,256 (0.17%)	3.740 (2.875; 4.866)	0.0% (0.0; 89.6)	0.000 (0.000; 0.731)	0.79 (0.675)
LRTD (3+ symptoms)	[48,61,85]	42/33,130 (0.13%)	17/47,256 (0.04%)	4.326 (2.415; 7.748)	0.0% (0.0; 89.6)	0.000 (0.000; 6.872)	0.65 (0.721)

**Table 8 vaccines-12-00500-t008:** Summary of Egger’s test for publication bias in sampled studies.

Settings	Pathogen	t	df	*p*-Value	Bias (SE)
First season	ARI	1.46	3	0.241	4.819 (3.303)
	LRTD 2 symptoms	2.09	2	0.171	36.073 (17.219)
	LRTD 3+ symptoms	−1.70	2	0.232	−0.494 (0.291)
Follow-up	ARI	1.66	1	0.345	5.566 (3.348)
	LRTD 3+ symptoms	−2.76	1	0.222	−1.774 (0.644)

## Data Availability

Data are available on request to the corresponding author.

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
