# Peer review of "Efficacy of Respiratory Syncytial Virus Vaccination to Prevent Lower Respiratory Tract Illness in Older Adults: A Systematic Review and Meta-Analysis of Randomized Controlled Trials"

_vaccines, 2024, doi:10.3390/vaccines12050500_

Round 1

Reviewer 1 Report

Comments and Suggestions for Authors

This is a comprehansive and well-written systematic review and meta-analysis. My only comment is relatively minor in nature.

Much of the mnanuscript could be condensed without loss of key information. For example, the introduction section does not need detail about the structure of RSV; it can be assumed that readers will have some familiarity with the virus and associated infections. Reference to 63 publications in the introductory section is an indicator that a lot of this material is better placed in the discussion section. 

Author Response

Estimated Reviewer 1,

thank you in advance for your collaborative suggestions, whose implementation has substantially improved the overall quality of our study. We agreed about the high number of references included in the introduction. However, due to the very innovative topic of preventive measures against RSV infection we are confident about the significance and the importance of referencing the paper with evidence based data. However, where possible and consistent with recommendations from Rev.2 and Rev.3, we amended the text for reducing its length (e.g. by removing the comment about metapneumovirus that we agree as not consistent with the aim of the paper).

In summary, we again thank you for your valuable comments and contributions to our study.

On the behalf of all authors,

M. Riccò

Reviewer 2 Report

Comments and Suggestions for Authors

Thank you for the opportunity to review this manuscript. The manuscript's objective is to analyze a interesting topic: to evaluate vaccine efficacy of RSV vaccines in older adults, providing valuable evidence and possibly guidance for clinical use.

I have some suggestion to improve this manuscript.

The introduction should be expanded. In particular, It would be appropriate the authors to give an idea of the burden of RSV on older adults. In Industrialized Countries, RSV incidence rates and hospitalization rates among older adults are estimated to be 600 cases/100000 person-year "Kenmoe et al. Current Opinion in Infection Disease, April 2024). In addition it's important to addedd information about the vaccine. Two RSV vaccines are approved for adults ages 60 years and older: Arexvy and Abrysvo.

Methods are well structured with a rigorous methodology but I suggest to insert the rusult of quality assesment in table 1. In addition I suggest to insert the quality assessment table as supplememtal materials.

Author Response

Estimated Reviewer 2,

thank you in advance for your collaborative suggestions, whose implementation has substantially improved the overall quality of our study. In the following lines, we will reply point-to-point to your suggestions. Thank you again.

The introduction should be expanded. In particular, It would be appropriate the authors to give an idea of the burden of RSV on older adults. In Industrialized Countries, RSV incidence rates and hospitalization rates among older adults are estimated to be 600 cases/100000 person-year "Kenmoe et al. Current Opinion in Infection Disease, April 2024). In addition it's important to addedd information about the vaccine. Two RSV vaccines are approved for adults ages 60 years and older: Arexvy and Abrysvo.

Thank you for your comments.

We amended the introduction as follows, by including the suggested reference from Kenmoe and reporting more details on the vaccines.

According to available estimates, every year RSV causes around 3.5 million hospital admissions in infants aged 0 to 60 months [1,3], even without noticeable comorbidities [2,13,14], with high hospital admission rates) [15–19], recently estimated around 5.3 per 1,000 people (95% confidence interval [95%CI]) 4.2-6.8 at global level [1,3]. Nonetheless, with around 158,229 hospitalizations occurring annually in adults aged ≥ 18 years, 92% of them from adults aged ≥ 65 years [20], with a corresponding hospitalization rate of 157 per 100,000 [21], RSV also affects older individuals [13,22–24], being increasingly acknowl-edged as a cause of respiratory illnesses and severe low-respiratory tract infections in adults [10], mostly in older adults [25,26], where it causes high morbidity and mortality [24,27,28], particularly among institutionalized subjects [2,29–31], with a significant public health impact [32–37]. Until 2023, preventive and treatment options for older adults were limited, as several RSV candidate vaccines did ultimately fail to achieve targeted efficacy, that in children was initially identified in a greater than 70% vaccine efficacy (VE) against confirmed severe RSV disease over at least one year post vaccination, as well as the secondary target of 50% VE against confirmed severe RSV [38]. Since then, several vac-cines were able to complete phase III clinical trials, and two preventive vaccines were ultimately licensed for human use, Abrysvo from Pfizer Inc. (Pfizer Europe MA EEIG, Brussels, Belgium) and Arexvy from GlaxoSmithKline LLC (GlaxoSmithKline Biologicals SA, Rixensart, Belgium) [39,40]. Both of them are subunit vaccines based on pre-fusion F protein, but while Abrysvo is a divalent, non-adjuvated formulate, Arexvy is a mono-valent formulate based on the pre-fusion protein F of RSV from group A [39,40].

Methods are well structured with a rigorous methodology but I suggest to insert the rusult of quality assesment in table 1. In addition I suggest to insert the quality assessment table as supplememtal materials.

Thank you for your suggestions.

We’ve amended Table 1 and edited ROB materials as suggested.

In summary, we again thank you for your valuable comments and contributions to our study.

On the behalf of all authors,

M. Riccò

Reviewer 3 Report

Comments and Suggestions for Authors

I congratulate the authors, their analysis of the work regarding RSV vaccines is interesting, complete and very exhaustive, also considering the impact that RSV infections have in terms of costs for the healthcare system. I only have a few minor observations.

Minor questions:

1.     In the introduction section (lines 60-65) the authors describe how RSV causes a high rate of hospitalization in infants, I would suggest the authors give some numbers (for example 4/10000 etc.). The same below when the authors refer to the high morbidity and mortality in older adult subjects.

2.     In the introduction section the authors describe 2 vaccines licensed for use in humans to protect against RSV (lines 70-73). Abrysvo from Pfizer, and Arexvy from Glaxo. I would suggest the authors include a brief description of what type of vaccines they are (for example, viral antigen vaccine with carrier protein and adjuvants rather than mRNA vaccine or if live attenuated vaccine etc.)

3.     In the Results section in Table 1 (lines 216-217) the authors list the studies, the vaccine, and the time period in which the study was conducted. I would suggest that the authors add a column, in Table 1, where they can insert the category to which the vaccines considered belong (if mRNA, if antigens with carrier protein and adjuvants, etc.). I ask this because I consider it useful to report the type of vaccine used, as the mechanism of action is different depending on the type.

4.     In lines 308-310 of the results section the authors show the calculations for separate estimates for LRTD cases with 2 symptoms or with 3 or more symptoms (figure A6). To help the reader I would suggest the authors insert a line summarizing the type of symptoms they are talking about (a brief mention is enough).

5.     In the results section, the authors report an in-depth statistical evaluation of the efficacy of the different RSV vaccines taken into consideration in their meta-analysis. However, it would be interesting to add a brief description of the antibody levels produced after vaccination. If the antibody levels have been analyzed and reported (in the studies taken into consideration) this data should also be reported and perhaps subjected to some statistical inference, to make the work even more solid and complete.

6.     In the results section (lines 348-362) the authors describe the three studies that provided follow-up data. Here too I would ask the authors to indicate the main symptoms of LRTD described in the relevant studies, for completeness (half a line of description is enough).

Author Response

Estimated Reviewer 3,

thank you in advance for your collaborative suggestions, whose implementation has substantially improved the overall quality of our study. In the following lines, we will reply point-to-point to your suggestions. Thank you again.

In the introduction section (lines 60-65) the authors describe how RSV causes a high rate of hospitalization in infants, I would suggest the authors give some numbers (for example 4/10000 etc.). The same below when the authors refer to the high morbidity and mortality in older adult subjects.

Thank you for your suggestion. This section has been amended as follows:

According to available estimates, every year RSV causes around 3.5 million hospital admissions in infants aged 0 to 60 months [1,3], even without noticeable comorbidities [2,13,14], with high hospital admission rates) [15–19], recently estimated around 5.3 per 1,000 people (95% confidence interval [95%CI]) 4.2-6.8 at global level [1,3]. Nonetheless, with around 158,229 hospitalizations occurring annually in adults aged ≥ 18 years, 92% of them from adults aged ≥ 65 years [20], with a corresponding hospitalization rate of 157 per 100,000 [21], RSV also affects older individuals [13,22–24], being increasingly acknowl-edged as a cause of respiratory illnesses and severe low-respiratory tract infections in adults [10], mostly in older adults [25,26], where it causes high morbidity and mortality [24,27,28], particularly among institutionalized subjects [2,29–31], with a significant public health impact [32–37].

In the introduction section the authors describe 2 vaccines licensed for use in humans to protect against RSV (lines 70-73). Abrysvo from Pfizer, and Arexvy from Glaxo. I would suggest the authors include a brief description of what type of vaccines they are (for example, viral antigen vaccine with carrier protein and adjuvants rather than mRNA vaccine or if live attenuated vaccine etc.)

Thank you again. We agreed with your suggestions and the text was amended as follows:

Both of them are subunit vaccines based on pre-fusion F protein, but while Abrysvo is a divalent, non-adjuvated formulate, Arexvy is a monovalent formulate based on the pre-fusion protein F of RSV from group A [39,40].

Moreover, a specifically designed Table A3 includes some details on the vaccines

In the Results section in Table 1 (lines 216-217) the authors list the studies, the vaccine, and the time period in which the study was conducted. I would suggest that the authors add a column, in Table 1, where they can insert the category to which the vaccines considered belong (if mRNA, if antigens with carrier protein and adjuvants, etc.). I ask this because I consider it useful to report the type of vaccine used, as the mechanism of action is different depending on the type.

Thank you: Table 1 has been redrawn as requested. Moreover, we’ve implemented ROB assessment in the Table 1, as suggested by other reviewers, removing summary Figure and Annex Table with individual qualitative assessment  (as now included in Table 1).

In lines 308-310 of the results section the authors show the calculations for separate estimates for LRTD cases with 2 symptoms or with 3 or more symptoms (figure A6). To help the reader I would suggest the authors insert a line summarizing the type of symptoms they are talking about (a brief mention is enough).

Thank you. Corresponding section has been revised in sub-titles for assessed groups of patients. Moreover, we’ve specifically designed an Annex Table (Appendix Table A4) including the details of signs and symptoms included in case definitions, and the Table is cited across the main text.

(“Regarding LRTD, distinctive estimates were calculated for LRTD with 2 symptoms and with 3 or more symptoms (see Appendix Table A4 for details)”.

In the results section, the authors report an in-depth statistical evaluation of the efficacy of the different RSV vaccines taken into consideration in their meta-analysis. However, it would be interesting to add a brief description of the antibody levels produced after vaccination. If the antibody levels have been analyzed and reported (in the studies taken into consideration) this data should also be reported and perhaps subjected to some statistical inference, to make the work even more solid and complete.

Thank you for your suggetion. In fact, while we agree about the significance of this data in order to make the work more solide and complete, immunogenicity data are quite difficult to reconciliate with pooled estimates on Vaccine Efficacy, because of the heterogeneity of reporting and design of the source papers.

However, we implemented a newly designed Appendix Table A8 that is referenced within the main text as follows:

Second, even though most included studies were of appropriate or even of high quality, we gathered data on diverse vaccine strategies, and the variable vaccine type inquired in retrieved RCT reasonably introduced high heterogeneity in the results, potentially affecting the final results. Not coincidentally, estimates on immunogenicity of assessed vaccines were irregularly provided across the retrieved studies: their summary is provided in Appendix Table A8. In this regard, despite the differences in vaccine strategy and heterogeneity in reporting strategy, all included vaccines were documented as highly immunogenic, at least during the first months after the delivery of the vaccine.

In the results section (lines 348-362) the authors describe the three studies that provided follow-up data. Here too I would ask the authors to indicate the main symptoms of LRTD described in the relevant studies, for completeness (half a line of description is enough).

Again, thank you for your note. We referenced to the aforementioned Table A4 in the main text and modified the main text as follows:

The three studies providing data on the follow-up of vaccinated individuals [48,61,85] provided pooled estimates for VE regarding the occurrence of ARI and LRTD with 3 or more symptoms (see Annex Table A3 for case definition). More precisely, the VE on ARI was estimated to 46.64% (95%CI 35.94 to 55.55), and 61.15% (45.29 to 72.40) for LRTD with 3 or more symptoms. Individual estimates for ARI ranged between 40.04% (95%CI 18.90 to 55.67) for RSVPreF3, 41.38% (95%CI 16.97 and 58.62) for RSVpreF, and peaked to 53.69% (95%CI 40.24 to 64.11) for mRNA-1345, while estimates for LRTD with 3 or more symptoms was 55.83% (95%CI 28.41 to 72.75), 78.65% (95%CI 25.72 to 93.86), and 62.88% (95%CI 37.17 to 78.07) for RSVpreF3, RSVpreF, mRNA-1345 respectively (Annex Figure A17 and A18).

In summary, we again thank you for your valuable comments and contributions to our study.

On the behalf of all authors,

M. Riccò
